# Memory-efficient Segmentation of High-resolution Volumetric MicroCT Images

**Yuan Wang**[1]                                                      YW720@IMPERIAL.AC.UK

**Laura Blackie**[2]                                        LAURA.BLACKIE@LMS.MRC.AC.UK

**Irene Miguel-Aliaga**[2]                           I.MIGUEL-ALIAGA@LMS.MRC.AC.UK

**Wenjia Bai**[1,3]                                              W.BAI@IMPERIAL.AC.UK

[1] *Department of Computing, Imperial College London, London, UK*

[2] *Institute of Clinical Sciences, Imperial College London, London, UK*

[3] *Department of Brain Sciences, Imperial College London, London, UK*

**Editors:** Under Review for MIDL 2022

## Abstract

In recent years, 3D convolutional neural networks have become the dominant approach for volumetric medical image segmentation. However, compared to their 2D counterparts, 3D networks introduce substantially more training parameters and higher requirement for the GPU memory. This has become a major limiting factor for designing and training 3D networks for high-resolution volumetric images. In this work, we propose a novel memory-efficient network architecture for 3D high-resolution image segmentation. The network incorporates both global and local features via a two-stage U-net-based cascaded framework and at the first stage, a memory-efficient U-net (meU-net) is developed. The features learnt at the two stages are connected via post-concatenation, which further improves the information flow. The proposed segmentation method is evaluated on an ultra high-resolution microCT dataset with typically 250 million voxels per volume. Experiments show that it outperforms state-of-the-art 3D segmentation methods in terms of both segmentation accuracy and memory efficiency.

**Keywords:** 3D volumetric image segmentation, high-resolution image segmentation, network design, memory-efficient network.

## 1. Introduction

Although deep learning has demonstrated excellent segmentation performance in many areas, there are still several unmet challenges that inhibit its deployment in biomedical applications. For example, medical imaging data are often high-dimensional (3D or 4D) (Çiçek et al., 2016) with large image size (Zhu et al., 2018), where training is often limited by the GPU memory constraint. Mainstream methods perform patch-based training (Campanella et al., 2019)(Bui et al., 2019)(Yu et al., 2017) or image downsampling (Oktay et al., 2018) to deal with the large input and reduce memory consumption. However, both approaches have inherent drawbacks: patch cropping reduces the field of view of the neural network and thus loses global information, whereas image downsampling leads to the loss of fine-grained details. Another challenge in medical image segmentation is the class imbalance issue. Some label classes only take a small proportion of a high-resolution 3D image. Performing image downsampling, in particular with a large downsampling factor, can easily damage the crucial information for small-object localisation and result in inferior segmentation accuracy of under-represented classes.

The potential to improve segmentation performance by introducing novel network architectures is also limited with insufficient GPU memory. For example, residual networks (He et al., 2016), deep supervision (Lee et al., 2015) and the recently proposed Transformer (Chen et al., 2021) architectures typically do not consider the adaptation when training on high-resolution 3D images. Their extensions to 3D image segmentation may not work well because the size of inputs has to be vastly reduced to accommodate the heavy increase in GPU memory footprint.

To address the afore-mentioned challenges in segmenting high-resolution 3D medical images, we propose a memory-efficient U-net architecture. The hypothesis is that both large spatial context and detailed local information are beneficial to the segmentation performance. We present a novel cascaded framework that accumulates the spatial context of the proposed memory-efficient U-net and links the two stages using a novel cascaded strategy, named post-concatenation. Experimental results show that the proposed method achieves high performance on high-resolution class-imbalanced volumetric microCT images in terms of both segmentation accuracy and memory efficiency.

## 2. Related Work

**Network architectures:** The CNN-based encoder-decoder structure is widely used in medical image segmentation with great success. The architecture represented by U-net (Çiçek et al., 2016)(Ronneberger et al., 2015) has achieved state-of-the-art results in the segmentation of organs, such as the heart, lung and liver (Isensee et al., 2021)(Oktay et al., 2018). Built upon the U-net, (Milletari et al., 2016) proposed to take residual structure to facilitate the feature extraction. Zhu *et al.* (Zhu et al., 2017) applied deep supervision on both the encoder and the decoder to mitigate the information loss during training. (Zhou et al., 2018) proposed the U-net++ architecture and introduced densely nested skip connections to improve the gradient flow in learning. Those modifications enhance the performance while leading to a large memory consumption as more activations are generated. An alternative approach (Reich et al., 2021) proposed to leverage CNN encoder to learn decision boundary in a function space for 3D shapes representation. Despite reduced memory consumption, the approach comes with lower performance compared to pure CNN architecture that generates voxelised segmentations.

Recently, the Transformer-based architecture has shown excellent success (Dosovitskiy et al., 2021). A commonly adopted strategy for image segmentation is to take a hybrid CNN-Transformer-based architecture (Xie et al., 2021; Ji et al., 2021). (Chen et al., 2021) proposed TransUnet structure that embeds Transformer in the encoder to enhance the long-distance dependency in features for 2D image segmentation tasks. (Hatamizadeh et al., 2021) employed UnetR, which takes a pure Transformer as the encoder for 3D image segmentation. However, it is necessary to take a small patch size in the Transformer in order to obtain competitive performance, which results in a significant increase in memory consumption. Focusing on reducing the spatial complexity, (Xie et al., 2021) introduced deformable Transformer in CNN-based encoder-decoder architecture on volumetric data segmentation. However, the training still leads to an expensive memory footprint compared to CNN-based architecture.

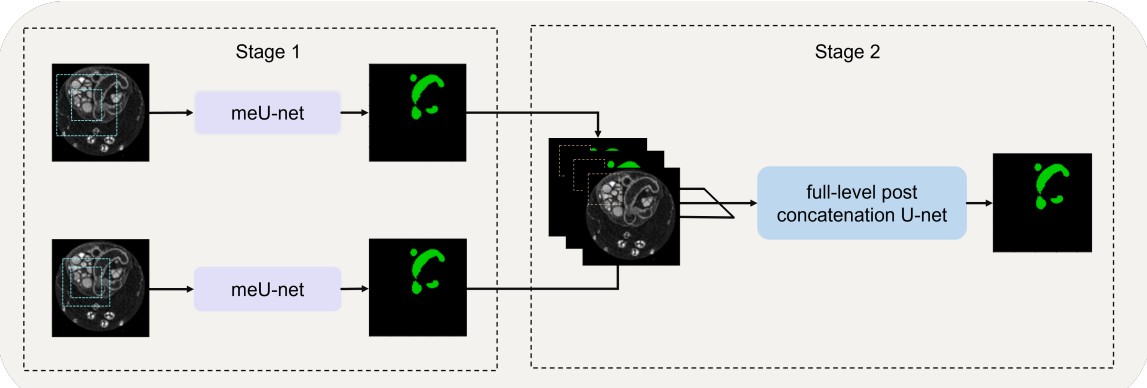

Figure 1: Proposed cascaded network. The first stage takes dual branches to extract spatial context at different resolution levels. Each branch utilises a memory-efficient U-net (meU-net) and samples two patches of different sizes for training. The second stage ensembles the first stage segmentation with standard U-net architecture to train a full-level post-concatenation U-net. Patches of standard size are sampled for training.

**Cascaded frameworks:** To improve segmentation performance, (Roth et al., 2018) proposed to cascade two single networks that integrate both global and local information. Training a cascaded network end-to-end improves performance while coming with a high cost of GPU memory (Jiang et al., 2019)(Liu et al., 2019). Single networks can also be trained separately in order to fit in the limited memory budget, which generates coarse prediction firstly and the result is refined in the following stage. In this regard, nnU-net (Isensee et al., 2021) proposed a 3D U-net-based cascaded network and demonstrated a great potential dealing with high-resolution datasets. Even though coming with great success, the class imbalance issue limits its application. The network implements the global spatial extraction through image downsampling in the first stage, which leads to a loss in details and texture information and thus may degrade the accuracy of small objects. Also, a large downsampling factor must be used at the first stage if the input images are extremely large. It is worth noting that the coarse segmentation with low quality from the first stage may misguide the second stage and lower the overall performance of the cascaded network.

## 3. Methods

In this section, we introduce the proposed cascaded network (see Figure 1). The first stage consists of memory-efficient U-net (meU-net) models to deal with large input patch size (see Figure 2). At the second stage, we ensemble a standard 3D U-net with the proposed post-concatenation method (see Figure 3). To train the proposed cascaded network, we take image patches of the original spatial resolution as input for both stages.

### 3.1. Stage 1: Memory-efficient U-net

We observe that the biggest bottleneck of GPU memory consumption for an encoder-decoder or U-net architecture lies in the first resolution level (Brügger et al., 2019). Without losing

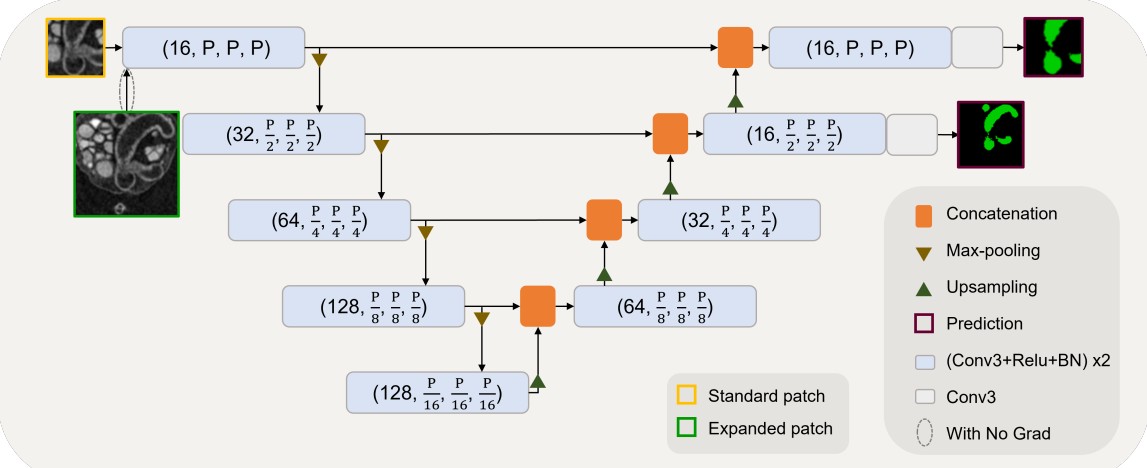

Figure 2: Illustration of the proposed meU-net. The size of standard patches is represented by $P \times P \times P$. The expanded patches of large size $(kP \times kP \times kP)$ supervise only the deep levels of the network so as to enable the model to capture a large field of view at a low memory cost. The feature maps generated from both the standard and expanded patches are uniformly represented by (P, P, P) in size at the highest level in the figure.

generality, let us denote an input image patch to the network is of size $P \times P \times P$ and each convolutional layer at the first level has $C$ channels. The memory consumption of the convolutional feature map is then proportional to $P^3 C$. If we use a larger image patch (expanded patch) of size $kP \times kP \times kP$ (k refers to patch expanding factor) to feed a more global context to the network, the memory consumption will increase by $k^3$ times. In common deep learning libraries such as PyTorch or Tensorflow, a gradient map of the same size as the feature map needs to be allocated during the model training phase, which further doubles the memory consumption.

To alleviate the memory footprint, we propose a memory-efficient U-net (meU-net) as illustrated in Figure 2. The meU-net supervises the network training with two input image patches. Patches of standard size $(P \times P \times P)$ are sampled firstly. We then sample expanded patches $(kP \times kP \times kP)$ to provide global context to the network, as shown in Figure 1. We implement patch expanding from the centre of the standard patches, with an extended sampling range along each axis. To save memory, the gradient calculation at the first resolution level is disabled for the expanded patches, which halves the memory consumption. Furthermore, the expanded patches only generate segmentation at the second resolution level of the decoder. Inspired by deep supervision (Xie and Tu, 2015), a loss function is defined at the second resolution level. The expanded patch will enable the update of all network parameters except the first level. To update the parameters at the first level of the network, the standard patch, which is the central region of the expanded patch, is used. The gradient calculation is enabled for the standard patch at the first level. On the decoder part, the standard patch will also generate a segmentation at the first level, where another loss function is defined.

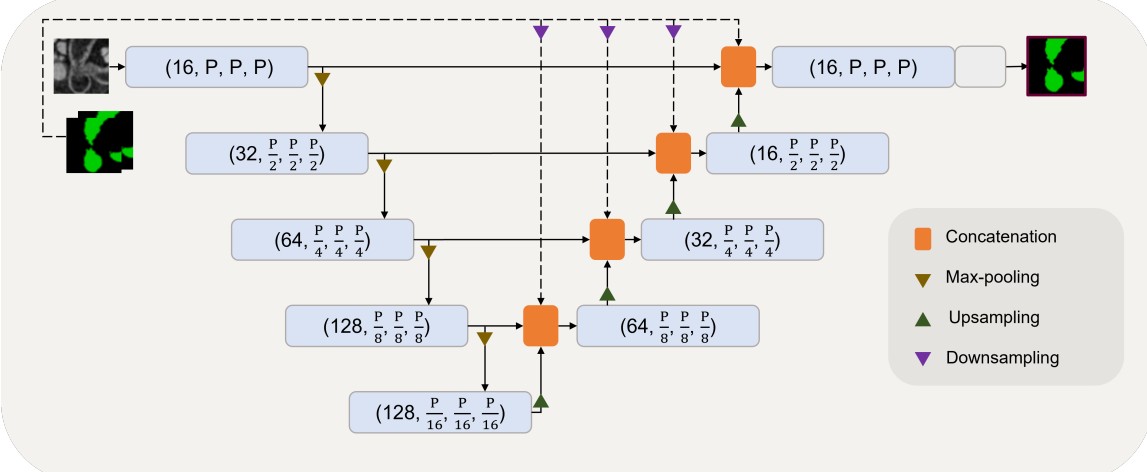

Figure 3: Illustration of proposed full-level post-concatenation U-net. We concatenate the first stage prediction with the model post the encoder. The first stage predictions supervise all decoder levels.

### 3.2. Fine-grained Cascaded Framework

The meU-net extracts global features at Stage 1, which are further concatenated to a Stage 2 network for final segmentation. As discussed above, cascaded networks commonly train a single model on downsampled images to obtain global context and subsequently, refine the coarse prediction by training on full-resolution patches. A key limitation of such architecture is that a heavy image downsampling must be implemented to meet the GPU requirement for high-resolution data. This comes at the cost of detailed information and leads to errors for small segmentation objects. We propose a novel cascaded network (see Figure 1) as an alternative solution. Major novelty can be divided into three categories: 1) employ the meU-net for spatial context extraction. 2) employ dual branches for multi-size training. 3) employ a novel post, full-level concatenation strategy for stage connection.

#### 3.2.1. Post concatenation

A common stage connection approach in cascaded networks is to take the first stage predictions as additional channels for the original image, and feed the concatenated input into the second-stage network (Isensee et al., 2021; Jiang et al., 2019). We argue that the fused input allows the encoder to overuse the coarse segmentation, as both the coarse predictions and raw images are processed simultaneously at the start of the encoder. The encoder may therefore simply utilise the coarse-grained predictions without updating the parameters heavily to learn extract features from raw images, which may lead to an easy decrease in loss during training, but comes with a limited accuracy improvement. We instead propose to fuse the extracted features with coarse-grained segmentation at the decoder level. This allows the encoder to bypass the interference when extracting features from the original images. We therefore employ the first stage segmentation to supervise all decoder levels,

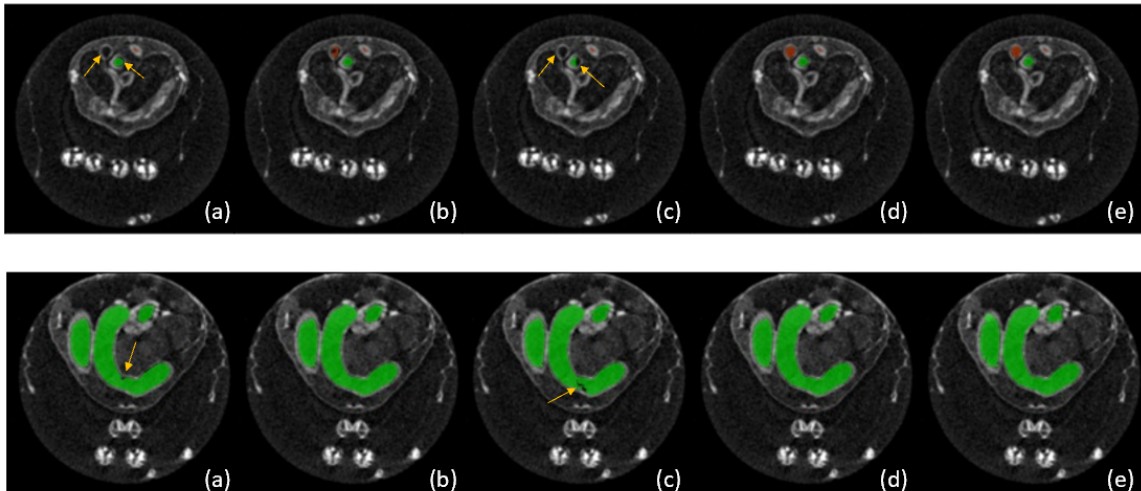

Figure 4: Comparison of the segmentation results on two exemplar subjects. The green part denotes the mid-gut and the red part denotes the hind-gut, which is typically very small. The yellow arrow highlights the segmentation error. (a) nnU-net; (b) proposed cascaded network (two-stage training using 280×280×280 patches and 160×160×160 patches); (c) 3D U-net; (d) Stage 1 meU-net (trained with 240×240×240 patches); (e) ground truth.

namely full-level post-concatenation, by downsampling the full-resolution predictions to the corresponding size in each level, as shown in Figure 3.

## 4. Experimental Results

### 4.1. Data

A dataset of 30 high-resolution 3D microCT images of the fruitfly is used. Each image is of size 1072×470×470 voxels and with a spatial resolution of 2.95×2.95×2.95 $\mu$m$^3$. There are two foreground classes, namely the mid-gut and hind-gut of the fruitfly, which are manually annotated by experts. The labels are imbalanced as the hind-gut is much smaller than the mid-gut (illustrated in Appendix B). The dataset is randomly split into 70%, 10% and 20% for training, validation and testing, respectively.

### 4.2. Experiments

We conducted experiments to evaluate 1) the memory efficiency of the proposed meU-net in Table 1. 2) the relationship between the patch expanding in meU-net with the GPU memory consumption in Table 2. 3) the performance of the proposed cascaded network compared to the previous state-of-the-art cascaded architecture in Table 3.

**meU-net** We first compare the proposed meU-net to a 3D U-net (Çiçek et al., 2016) in terms of GPU memory consumption and segmentation accuracy, the latter evaluated using the Dice score. Experiments are conducted with the same image patch size and batch size for a fair comparison in memory consumption. We set the patch expanding of 1.5× in each

| Architecture | Image downsampling | Patch size | Batch size | GPU memory(GB) | with data aug. | | w/o data aug | |
|---|---|---|---|---|---|---|---|---|
| | | | | | mid-gut | hind-gut | mid-gut | hind-gut |
| U-net | ↓2 | 240×240×240 | 1 | 8 | 0.921 | 0.581 | 0.913 | 0.563 |
| meU-net | ↓2 | 240×240×240 | 1 | 4 | 0.923 | 0.560 | 0.910 | 0.579 |
| U-net | - | 160×160×160 | 4 | 10.5 | 0.955 | 0.645 | 0.938 | 0.672 |
| meU-net | - | 160×160×160 | 4 | 6 | **0.958** | 0.634 | **0.942** | 0.654 |
| U-net | - | 240×240×240 | 4 | 30* | - | - | - | - |
| meU-net | - | 240×240×240 | 4 | 13 | 0.932 | **0.721** | 0.940 | **0.741** |

Table 1: Experiments for single-stage architecture. Comparison on the fruitfly dataset in terms of Dice metrics and memory consumption. me-Unet expands patches 1.5× in each axis by default. * represents the estimated memory consumption.

| Architecture | Patch expanding | Patch size | Batch size | GPU memory(GB) | with data aug. | | w/o data aug | |
|---|---|---|---|---|---|---|---|---|
| | | | | | mid-gut | hind-gut | mid-gut | hind-gut |
| meU-net | 1.00 | 160×160×160 | 4 | 10.5 | 0.952 | 0.632 | **0.953** | 0.666 |
| | ↑1.25 | 200×200×200 | 4 | 12 | **0.953** | 0.713 | 0.938 | 0.740 |
| | ↑1.50 | 240×240×240 | 4 | 13 | 0.932 | **0.721** | 0.940 | 0.741 |
| | ↑1.75 | 280×280×280 | 4 | 15 | 0.941 | 0.718 | 0.941 | **0.748** |

Table 2: Ablation study for patch expanding. We take standard patches of size 160×160×160 as the centre to sample larger patches. The standard patches supervise the entire network. The expanded patches supervise the deep network levels.

axis by default when training meU-net models, to get a balance between spatial context and training time.

Table 1 reports memory consumption and segmentation accuracy of meU-net, compared to standard U-net, given the same patch size and batch size. In general, meU-net can reduce the GPU memory consumption by 50% or more compared to U-net. Despite largely reduced memory (e.g. from 8GB to 4GB), meU-net achieved competitive results in Dice metrics. meU-net also allows us to train with larger patch sizes. For example, we are no able to train with patch size of 240×240×240 and this improves the Dice score from 0.654 to 0.741 on the hind-gut compared to training with patch size of 160×160×160. With the proposed meU-net, the memory required reduces from the expected 30GB to 13GB, and therefore enables improvement in prediction accuracy with a limited memory budget. The memory reduction is performed at the network training phase, which is the main bottleneck for developing 3D medical image segmentation networks.

Patch expanding in meU-net introduces additional spatial context and memory consumption. In Table 2, we evaluate the corresponding effect towards the Dice scores and GPU memory. The proposed meU-net is based on the hypothesis that having a large amount of spatial context information benefits segmentation performance. To test the hypothesis, we train a meU-net with an expanding factor of 1 (i.e., expanded patch = standard patch). The results show that meU-net achieved a similar performance compared to the standard U-net equivalent (Table 1, line 3) in terms of Dice score, when the same patch and batch size are used. However, we observe that by introducing extra context (1.95× in patch size, expanding patches 1.25× in each axis) to the standard U-net, the performance can be improved by more than 5% in terms of Dice score overall. We do not observe significant improvements in Dice scores when patches are highly expanded. We analyse that

| Architecture | Stage 1 | | Stage 2 | | GPU | Dice score | |
|---|---|---|---|---|---|---|---|
| | Patch size | Batch size | Patch size | Batch size | memory(GB) | mid-gut | hind-gut |
| nnU-net (Isensee et al., 2021) | 536×235×235 | 1 | 160×160×160 | 4 | 15.5 | 0.931 | 0.659 |
| proposed cascaded network | 280×280×280 | 4 | 160×160×160 | 4 | 15 | **0.948** | **0.768** |

Table 3: Comparison on two-stage cascaded networks. nnU-net takes two 3D U-net models in two stages. The first stage network trains on downsampled images. The proposed cascaded network takes meU-net in the first stage and full-level post-concatenation U-net in the second stage. For the proposed cascaded network, both stages take image patches of original spatial resolution as input.

the expanded patches of size 200×200×200 can largely cover the entire foreground in our case, while the further expanded patches mainly introduce the background region. Despite this, by introducing extra spatial context to 3D U-net, the patch expanding consistently improves prediction accuracy compared to the standard U-net across different expanding degrees.

We would like to note that a heavier patch expanding allows a larger memory saving. Training on patches of size 3.3× expanded (i.e., 1.5×1.5×1.5), the required memory increases to 1.24× compared to the standard U-net equivalent. With a much larger patch expanding 5.35× (i.e., 1.75×1.75×1.75), the memory requirement shows an increase to only 1.43×. A large patch expanding reduces the memory required heavily, and enables the model to train on image patches of large size, when the GPU memory is very limited.

**Cascaded network** We evaluate the performance of cascaded networks. The state-of-the-art nnU-net (Isensee et al., 2021) and our proposed cascaded network (meU-net + post-concatenation U-net) were trained for comparison. The detailed ablation study for proposed post-concatenation strategy can be found in Appendix A.9. Figure 4 illustrates exemplar segmentations and Table 3 reports quantitative results. We observe decreased Dice scores of nnU-net on both mid-gut and hind-gut (from 0.938 to 0.931 and from 0.672 to 0.659, respectively) compared to single-stage training (i.e., employ a single 3D U-net only). This evidenced our argument that the cascaded architecture with image downsampling in the first stage is suboptimal if dealing with high-resolution class-imbalanced images (see Section 2). In contrast, the results from Table 3 show the proposed cascaded network achieved improvements by a considerable margin, compared to both the single-stage model (e.g., meU-net and U-net) and previous state-of-the-art cascaded network (increased from 0.659 to 0.768 on hind-gut), with a similar memory requirement.

## 5. Conclusion

Here, we propose a simple but memory-efficient network architecture (meU-net) for high-resolution volumetric image segmentation. In conjunction with a novel stage connection strategy (post concatenation), the proposed cascaded network achieved an improved performance compared to the state-of-the-art segmentation networks and with a lower GPU memory budget. We expect the methods to be generalisable to other high-resolution image analysis tasks and lower the GPU memory cost in developing 3D segmentation networks.

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

## Appendix A. Implemention Details

### A.1. Data Preprocessing

The raw images were firstly cropped to be of size 1072x470x470. The nearest neighbour interpolation and bspline interpolation are used for label downsampling and data downsampling, respectively. We implement a 'sliding window' approach with overlapping to sample data. To balance the foreground with the background, we first set a threshold to do the filtering, followed by a roulette check that determines whether to accept the patch with pre-defined probability with the aim to help the model generalise on the background.

### A.2. Data Augmentation

We use commonly adopted affine transformation, including random rotation and random scaling. Although the performance of the mid-gut class reports a general improvement compared to the training without data augmentation, we do not observe an overall improvement due to the decreased performance in the hind-gut class. We therefore train the cascaded network without data augmentation to reduce the training time.

### A.3. Network Architecture

All of our works are built upon a unified 3D U-net. The number of channels at the end of each level of the encoder and decoder is (16,32,64,128,128) and (16,16,32,64), respectively. The model takes batch normalisation, max-pooling, and nearest-neighbour interpolation to implement feature map normalisation, downsampling, and upsampling.

### A.4. Loss Function

We employ the mainstream method (Sudre et al., 2017) which trains with weighted cross-entropy and soft Dice loss to deal with class imbalance issues. We take a softmax-like function (see Equation (1)) based on the counted number of voxels in each class to smooth the weight. The final Dice loss is the average of the loss in each class (see Equation (2)).

$$weight_i = softmax(\frac{\sum count}{count_i}) \qquad (1)$$

$$L_{\text{dice}} = 1 - \frac{2 * \sum L * P}{\sum L + \sum P + \epsilon} \qquad (2)$$

### A.5. Fine-grained Cascaded Network

We propose a novel stage connection approach for cascaded networks, which is the so-called **full-level post-concatenation** method. A detailed illustration can be found in Figure 3. In the implementation, the stage 1 predictions in the one-hot form are concatenated with the feature maps at the beginning of all decoder levels. We take nearest-neighbour interpolation to downsample the stage 1 predictions into corresponding resolutions.

The cascaded network is trained with dual meU-net modules in the first stage. The two branches take a patch expanding factor of 1.5x and 1.75x, respectively. For each meU-net, the network takes two patches as input. We still take the proposed sampling strategy as usual but further take the sampled patch as the centre to expand the sampling range (see Figures 1 and 2) and finally get a patch in large size. The sampled large patch is used for training the network layers that process low-resolution feature maps, while the small patch updates the whole network. To alleviate memory consumption, we omit the gradient calculation of the large patch at the highest level of the encoder. A new convolution operation is added at the decoder's second level, followed by a combined loss function (Dice loss + weighted cross-entropy) consistent with the one used in the highest level.

### A.6. Inference

We employ overlapping-based patch sampling for both training and inference. To do the fusion in inference, a weight-based strategy has been used to double weights the centre part of the patch as the edge of the patch lacks spatial context and thus comes with poor accuracy. The final prediction in each voxel is voted by overall weight, which is aggregated from overlapped patches.

Downsampling-based training requires an upsampling module to recover the original resolution. We take nearest-neighbour interpolation by default during inference.

### A.7. Training Procedure

We use Automatic Mixed-precision Training (AMP) to save GPU memory. When testing the GPU memory footprint, checkpointing is employed by default. Turning off checkpointing helps reduce training time but comes with larger memory consumption. We optimise the code and call the garbage collection manually once the intermediate variables are no longer needed (e.g., delete the input variable before the loss calculation and delete the loss variable

| Architecture | Image downsampling | Patch size | Batch size | GPU memory(GB) | with data aug. | | w/o data aug | |
|---|---|---|---|---|---|---|---|---|
| | | | | | mid-gut | hind-gut | mid-gut | hind-gut |
| U-net | ↓2 | 192×192×192 | 1 | 4 | 0.914 | 0.531 | 0.902 | 0.523 |
| meU-net | ↓2 | 240×240×240 | 1 | 4 | 0.923 | 0.560 | 0.910 | 0.579 |
| U-net | - | 128×128×128 | 4 | 6 | 0.933 | 0.632 | 0.935 | 0.641 |
| meU-net | - | 160×160×160 | 4 | 6 | **0.958** | 0.634 | **0.942** | 0.654 |
| U-net | - | 176×176×176 | 4 | 13 | 0.931 | 0.693 | 0.937 | 0.689 |
| meU-net | - | 240×240×240 | 4 | 13 | 0.932 | **0.721** | 0.940 | **0.741** |

Table 4: Comparison between U-net and meU-net when the same GPU memory is used. meU-net expands the patch size by a factor of 1.5 along each dimension.

| Concatenation | Stage 1 | | Stage 2 | | GPU memory(GB) | Dice score | |
|---|---|---|---|---|---|---|---|
| | Patch size | Batch size | Patch size | Batch size | | mid-gut | hind-gut |
| pre | 280×280×280 | 4 | 160×160×160 | 4 | 15 | 0.947 | 0.756 |
| post, full-level | 280×280×280 | 4 | 160×160×160 | 4 | 15 | **0.948** | **0.768** |

Table 5: Ablation study for concatenation method of proposed cascaded network.

before parameter update). Adam is used with a learning rate equal to 9e-4. The training will stop if the performance report no improvement after 30 epochs. In terms of patch size, we sample the patch with shape (1x160x160x160) by default and step 80 (half overlapping), with zero padding to ensure the entire image can be processed. Experiments are conducted on Nvidia P100 GPU with 16 GB memory available. The entire training for the proposed cascaded network takes more than eight days.

### A.8. Comparison for Single Model Given the Same GPU Memory

In the previous section, we evaluated proposed meU-net and standard U-net with the same patch and batch size (see Table 1). The meU-net reports competitive performance compared to standard U-net with largely reduced memory consumption. We further conducted experiments to make the comparison in terms of Dice score given the same memory consumption, as shown in Table 4. Experiments show that the meU-net achieved a general increase in performance due to a much larger patch size than the standard U-net, while with the same memory requirement.

### A.9. Ablation Study for Concatenation Method

To verify the proposed full-level post-concatenation strategy in the cascaded network, we conducted an ablation study, and the results are given in Table 5. The results show that the proposed post-concatenation method achieved better performance compared to the pre-concatenation strategy.

## Appendix B. Segmentation Dataset

Here we illustrate an example of our high-resolution imbalanced volumetric fruitfly images. Best viewed in colour.

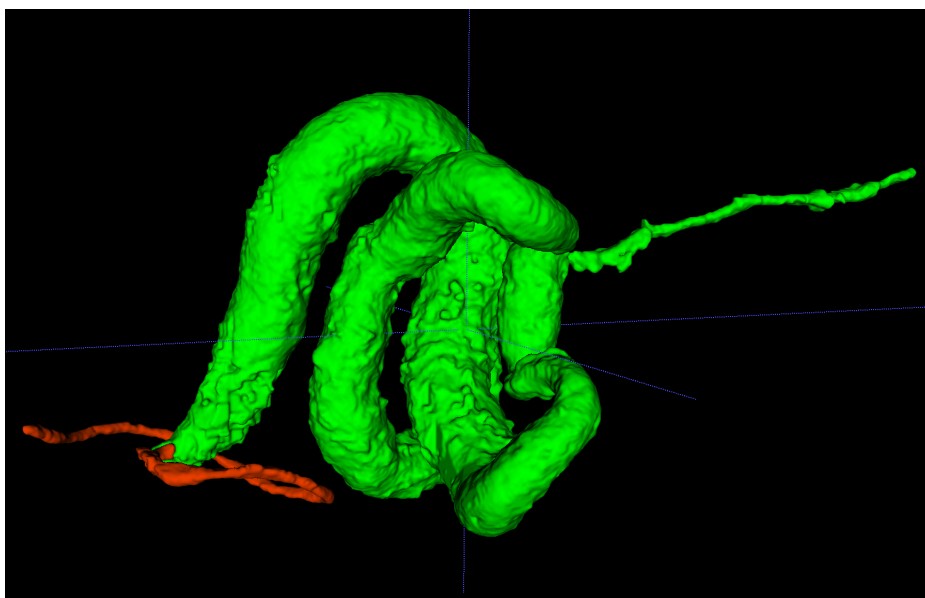

Figure 5: Example of fruitfly image in 3D format, with a focus on the foreground. A large part of the background region was cropped for a clear illustration. The green part represents the mid-gut, the red part represents the hind-gut. A heavy imbalance in foreground classes can be observed.

