# OpenReview forum: "Memory-efficient Segmentation of Volumetric High-resolution MicroCT Images"
_MIDL.io/2022/Conference — MIDL 2022_

### Official Review · Reviewer_sxSm · 2022-01-17

**Confidence:** 4
**Preliminary Rating:** 5
**Recommendation:** Oral

**Summary:**

In this paper, the authors proposed a novel and interesting method for memory-efficient segmentation of high-resolution 3D images. The proposed method is a two-stage cascaded framework to incorporate both global and local features. Experimental results show that the proposed method outperforms state-of-the-art segmentation methods in terms of both segmentation accuracy and memory efficiency.

**Strengths:**

- The authors designed a novel and interesting method for memory-efficient segmentation of high-resolution 3D images.  The method should be easily applicable for other segmentation tasks.
- The paper is well-written with detailed descriptions in Appendix.
- The experiments results are significance to demonstrate the effectiveness of proposed method with comparison to state-of-the-art segmentation methods.

**Weaknesses:**

- The experimental results show that “A large patch expanding reduces the memory required heavily, and enables the model to training on image patches of large size”, it would be interesting to see the comparison (larger patch size with smaller patch expanding v.s. smaller patch size with larger patch expanding).
- Do label1 and label2 in the table represent mid-gut and hind-gut respectively? Maybe clearer description should be given.


**Deanonymize Review:**

yes

**Paper Type:**

methodological development

**Questions To Address In The Rebuttal:**

- The experimental results show that “A large patch expanding reduces the memory required heavily, and enables the model to training on image patches of large size”, it would be interesting to see the comparison (larger patch size with smaller patch expanding v.s. smaller patch size with larger patch expanding).
- Do label1 and label2 in the table represent mid-gut and hind-gut respectively? Maybe clearer description should be given.


**Special Issue:**

yes

---

### Official Review · Reviewer_g9xY · 2022-01-24

**Confidence:** 4
**Preliminary Rating:** 3
**Recommendation:** Poster

**Summary:**

The authors introduce a change to the common U-Net to make the execution more memory-efficient.
By creating two unets, one of which runs on a higher resolution, but not calculating the gradient of the first layer on the full input but on a reduced input, roughly 50% less memory is used, which is especially relevant on 3d segmentation tasks were memory bounds are easily met.

**Strengths:**

The authors tackle a practical problem to reduce memory for 3d segmentation tasks, which is a real problem (talking from industry experience) and also present several innovations in the network architecture. On the (limited) comparison the new architecture performs well compared to a regular UNet and still reduces memory.

**Weaknesses:**

The change the authors propose is slightly complicated with three changes together:
Two 'reduced' unets, a cascade network with several changes, and a post concatenation network.
This adds questions to what contributes how much to the networks performance, and also raises questions like: What if we would keep a regular UNet the same, but just not compute the full gradient of the first network. This would be the simpler approach and should be compared against to see if the other changes are needed.

The authors do run an ablation study, but it is limited to comparing patch exanding factors and don't really do ablation (although it is still an appreciated study).

The main memory usage reduction seems to come from not computing the first layer gradients of the higher-resolution network.
This means the reduction is only relevant for training, and not execution time. This should be clarified, and reduces the usefulness in application cases.

Finally, the experiments compare the performance on a very reduced and seemingly not standard dataset. And the cases where the proposed network is better, the improvements are small (without error bars). This raises the question how reliable this result is, or if the changes are domain dependent.

**Deanonymize Review:**

yes

**Detailed Comments:**

The paper introduces several interesting changes to the UNet, with memory usage during training in mind. The changes are interesting and seem novel, however given the number of changes and the limited comparisons, it is hard to say how performant the network is for general UNet. Ideally a strong ablation study on more commonly used datasets is preferred.
That being said, the changes are interesting and the reduction in memory usage is useful and can inspire other work.

**Final Rating After The Rebuttal:**

4: Weak Accept

**Justification Of The Final Rating:**

The authors addressed my questions. I appreciate the value of any methods reducing memory consumption during training and keeping the same performance. Also the innovation of using two levels of detail, but using different backpropagation methods for them is interesting.
The main weak point of the paper is the validation on only one dataset, and it's not a standard benchmark. This makes the validation relatively weak.

**Paper Type:**

methodological development

**Questions To Address In The Rebuttal:**

Right now I'm borderline due to the limited comparison and relatively complicated network architecture, compared to the improvements in performance. However, I find the changes interesting and perhaps another experiment would help prove that the performance improvements are steady.

**Special Issue:**

no

---

### Official Review · Reviewer_gZbS · 2022-01-24

**Confidence:** 4
**Preliminary Rating:** 2

**Summary:**

The paper proposes a memory-efficient network architecture (meU-net) for high-resolution volumetric image segmentation. The authors claim that the proposed meU-net can achieve improved performance with lower GPU memory consumption compared to other state-of-the-art methods. The results were evaluated on fruitfly dataset.


**Strengths:**

1. The paper points out an important problem that in volumetric image segmentation, the modern GPU memory limits the application of deep neural networks.
2. The proposed meU-net achieved slightly better segmentation accuracy with lower GPU memory on fruitfly dataset.
3. The proposed method is easy to understand and implement, providing benefits for those people with limited GPU resources.


**Weaknesses:**

1. The method is not novel.
2. The method is only evaluated on one dataset, it would be nice to have different datasets across different scanners such as CT, MRI, ultrasound to verify the efficiency  and effectiveness of the proposed method.
3. There are multiple changes in the meU-Net compared to U-Net, the authors are encouraged to do an ablation study to verify the effectiveness of each component.


**Deanonymize Review:**

no

**Final Rating After The Rebuttal:**

3: Borderline

**Justification Of The Final Rating:**

The authors of the paper have addressed most of my concerns, thus, I would raise my rating to borderline.
However, neither disabling gradients at certain components nor applying post-encoder concatenation is sufficient novel.

**Paper Type:**

validation/application paper

**Questions To Address In The Rebuttal:**

1. Suppose U-Net and meU-Net using the same GPU memory, how is the performance comparison?
2. In Table 1, it seems that meU-net is better than its counterpart in dice of one of the two labels, but worse in dice of another label. I would say this is the trade-off rather than improvements, can the authors clarify on this?


**Special Issue:**

no

---

### Meta-Review · Area_Chair_6z22 · 2022-02-13

**Recommendation:** Accept (Oral)
**Confidence:** 5

**Metareview:**

The authors present a memory-efficient segmentation method of 3D images. The proposed method is a two-stage cascaded framework to incorporate both global and local features.  To alleviate the memory, one of the significant feature of the method is disable the gradient calculation at the first resolution level for large-size patches. Results are provided on a 30 high-resolution 3D microCT images of the fruitfly.

In the first round of review:
- All reviewers have acknowledged the great impact of such paper, as 3D image segmentation is a true practical problem
In addition, the method is well explained (ie good implementability) and the presented results are somehow good.
- BUT: The major issue was that experiments were reduced  to really convince the reader of the added value of the method. Additional comparative experiments and ablation studies were required.

The authors  were able to provide several additional experiments to deepen the understanding of the method and highlight its practical interest.

Based on the overall positive opinions of the reviewers at the 2nd stage of reviews, and the general interest of the method, I recommend to have it accepted as an oral contribution.

NB: note that this paper is the top of my stack, hence I was drawn to put it as an 'Oral' paper. In case too many papers are Oral candidates, this paper could reasonably be downgraded as a poster.

---

### Decision · Program_Chairs · 2022-02-28

Accept